# Superposition disentanglement of neural representations reveals hidden alignment

**André Longon**
UC San Diego
alongon@ucsd.edu

**David Klindt**
Cold Spring Harbor Laboratory

**Meenakshi Khosla**
UC San Diego

## Abstract

The superposition hypothesis states that single neurons may participate in representing multiple features in order for the neural network to represent more features than it has neurons. In neuroscience and AI, representational alignment metrics measure the extent to which different deep neural networks (DNNs) or brains represent similar information. In this work, we explore a critical question: *does superposition interact with alignment metrics in any undesirable way?* We hypothesize that models which represent the same features in *different superposition arrangements*, i.e., their neurons have different linear combinations of the features, will interfere with predictive mapping metrics (semi-matching, soft-matching, linear regression), producing lower alignment than expected. We develop a theory for how permutation metrics are dependent on superposition arrangements. This is tested by training sparse autoencoders (SAEs) to disentangle superposition in toy models, where alignment scores are shown to typically increase when a model's base neurons are replaced with its sparse overcomplete latent codes. We find similar increases for DNN→DNN and DNN→brain linear regression alignment in the visual domain. Our results suggest that superposition disentanglement is necessary for mapping metrics to uncover the true representational alignment between neural networks.

## 1 Introduction

Deep neural networks (DNNs) now possess highly competent behaviors in linguistic and perceptual domains, motivating researchers to understand their inner workings through mechanistic interpretability (MI), both for scientific insight and to ensure safe deployment [46]. A parallel discovery has been that these highly competent DNNs, despite being trained purely on task performance, spontaneously develop representations that closely resemble those in visual and language regions of primate brains [51, 21, 23, 45]. This latter discovery initiated the study of *alignment metrics* to capture the representational similarities between DNNs and brains. Popular metrics include representational similarity analysis (RSA) [31] which measures the similarity in neural geometries (stimulus-by-stimulus representational similarity matrices), and predictive mapping (semi-matching, soft-matching, linear regression, among others) which learns to predict brain responses from a DNN's neural responses. With such metrics serving as benchmarks, the field of NeuroAI strives to produce more brain-like DNNs [44, 32] and to understand which factors of the DNN, such as the architecture, training objective, or data diet, drive this representational convergence [19, 29, 8, 11]. Alignment metrics are also adopted to assess universality [30, 34, 2, 39, 16, 49, 7, 18], the occurrence of the same representations across neural networks, important to understand the principles of neural representations.

A major obstacle in the study of neural representations in DNNs is *polysemanticity* [39] (see mixed selectivity [43] for its treatment in neuroscience): when individual neurons appear to represent a confluence of seemingly unrelated features (e.g., a visual neuron which strongly responds to parts of both cats and cars), rendering them relatively inscrutable. This observation led MI to develop

the *superposition hypothesis* [13], which postulates that neurons represent multiple features so that the number of features a neural network can represent greatly surpasses its number of neurons. If features are sufficiently sparse (not all images contain cats or cars), then a subset of features which rarely co-occur may all be assigned to a single neuron, with features being distributed across neurons to form unique population codes. The superposition hypothesis in turn catalyzed the use of dictionary learning, especially sparse autoencoders (SAEs) [9, 3], to disentangle features from superposition by mapping their neural activity into a sparse overcomplete latent space. The dimensions of this space are often far more interpretable than the base neurons, thus seeming to correspond to individual features. The success of dictionary learning in the extraction of interpretable population codes has shifted the representational unit of analysis away from individual neurons [48, 49, 28, 27].

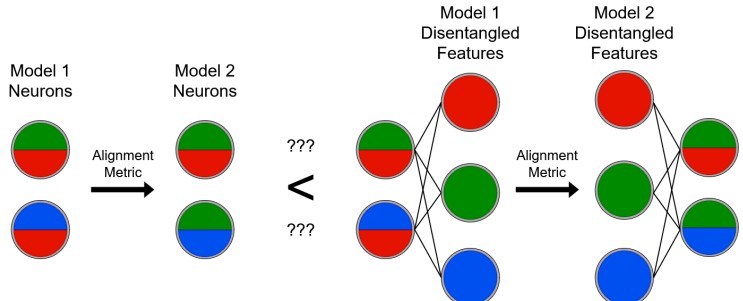

Figure 1: A visual depiction of the question: does superposition disentanglement increase representational alignment? Features are represented as colors, and neurons may arrange multiple features in superposition. The left side of the inequality shows an alignment metric taken between the base neurons of two models which represent the same features but in different superposition arrangements. The right side shows each model with their superimposed representations disentangled via a projection into an ideal sparse overcomplete space, where each dimension represents an individual feature. The same alignment metric is taken over these sparse latent codes in lieu of the base neurons.

Given that superposition and its disentanglement has transformed how MI researchers analyze model internals, a significant question for NeuroAI is raised: *does superposition interfere with representational alignment metrics?* Depicted in Figure 1, we hypothesize that if two models represent the same features in different superposition arrangements, this will deflate their alignment scores from predictive mapping.

First presented is a theory for how the more stringent metric of soft-matching [22] is deflated by different superposition arrangements. To empirically test this framework, toy models are trained from different random initializations to demonstrate that different superposition arrangements of the same features indeed occur. We then train SAEs on the toy models' neural activations and find that the soft-matching score between the two models sharply increases when their respective SAE latents are used in place of their base neurons. Next, identical DNNs are trained on ImageNet from different random initializations. After training SAEs on them, we find alignment increases in deep layers consistent with the toy models. Lastly, linear regression is investigated, the most flexible and widely used mapping metric. Alignment scores from regression also increase across the toy models and DNNs, but only when the source model's SAE latents are mapped to the target model's base neurons. A linear alignment increase is also found between DNNs and brains using the Natural Scenes Dataset (NSD) [1], a large dataset of human voxel responses to natural images. Insofar as the information we want from our mapping metrics is the degree of shared features between two neural networks, our results collectively suggest that superposition disentanglement is necessary to obtain this.

## 2   Alignment Metrics

To limit the scope, we focus on representational alignment metrics that are sensitive to the *representational form*—how information is implemented at the level of individual neurons. Because they depend on the representational basis, they are particularly relevant to questions of superposition, where the same features may be embedded in different basis arrangements across models. We employ the soft-matching correlation score [22], a symmetric generalization of semi-matching (aka pairwise) grounded in optimal transport. Here we provide a formal definition for the soft-matching metric (see

semi-matching in the appendix A.1). For all of our alignment experiments, we report the mean score and standard error across k=5 cross-validation folds.

**Soft-Matching Correlation Score**   Suppose we have response matrices from two neural networks, $\mathbf{Y}_a \in \mathbb{R}^{M \times N_a}$ and $\mathbf{Y}_b \in \mathbb{R}^{M \times N_b}$, containing the activity of $N_a$ and $N_b$ units respectively for $M$ stimuli. Let $\mathbf{P} \in \mathbb{R}^{N_a \times N_b}$ be a nonnegative matrix lying in the transportation polytope:

$$\mathcal{T}(N_a, N_b) = \left\{ \mathbf{P} \,\middle|\, \mathbf{P}\mathbf{1}_{N_b} = \tfrac{1}{N_a}\mathbf{1}_{N_a}, \ \mathbf{P}^\top\mathbf{1}_{N_a} = \tfrac{1}{N_b}\mathbf{1}_{N_b}, \ \mathbf{P} \geq 0 \right\}.$$

Here, each row sums to $1/N_a$ and each column to $1/N_b$, yielding a "soft permutation" that distributes mass across multiple matches.

If the columns of $\mathbf{Y}_a$ and $\mathbf{Y}_b$ are mean-centered and normalized, their dot product $\mathbf{y}_{a,i}^\top\mathbf{y}_{b,j}$ equals the correlation between units $i$ and $j$. The soft-matching correlation score is then defined as:

$$\text{Score}_{\text{softmatch}}(\mathbf{Y}_a, \mathbf{Y}_b) = \max_{\mathbf{P} \in \mathcal{T}(N_a, N_b)} \sum_{i=1}^{N_a} \sum_{j=1}^{N_b} \mathbf{P}_{ij} \, \mathbf{y}_{a,i}^\top\mathbf{y}_{b,j}. \tag{1}$$

When $N_a = N_b$, this reduces to finding the best one-to-one matching between units, equivalent to permutation-based alignment [50]. In general, it provides a symmetric and well-defined alignment metric even when the two neural populations differ in size.

## 3   Permutation-based Metrics Under Superposition

Let $M$ be the number of stimuli and $N$ the number of observed units in each representation. Let $F$ denote the number of latent features (typically $F \geq N$). We assume a shared latent generator across systems: the $M \times F$ matrix $\mathbf{Z}$ stacks latent feature vectors for the $M$ stimuli as rows, with each $\mathbf{z}$ being $K$-sparse, i.e., $\|\mathbf{z}\|_0 \leq K$.

Two observed representations are linear superpositions of the same latent features:

$$\mathbf{Y}_a = \mathbf{Z}\mathbf{A}_a, \qquad \mathbf{Y}_b = \mathbf{Z}\mathbf{A}_b, \qquad \mathbf{A}_a, \mathbf{A}_b \in \mathbb{R}^{F \times N}.$$

Column $i$ of $\mathbf{A}_\bullet$ contains the latent mixing weights for unit $i$ in system $\bullet \in \{a, b\}$. Superposition means there are more features than neurons $F > N$.

For any column vector $\mathbf{u} \in \mathbb{R}^M$, write $\langle \mathbf{u}, \mathbf{v} \rangle_M := \frac{1}{M}\mathbf{u}^\top\mathbf{v}$ and $\|\mathbf{u}\|_M := \sqrt{\langle \mathbf{u}, \mathbf{u} \rangle_M}$. Define the (sample) Pearson correlation between columns $\mathbf{y}_{a,i}$ and $\mathbf{y}_{b,j}$ by

$$\rho(\mathbf{y}_{a,i}, \mathbf{y}_{b,j}) = \frac{\langle \mathbf{y}_{a,i}, \mathbf{y}_{b,j} \rangle_M}{\|\mathbf{y}_{a,i}\|_M \, \|\mathbf{y}_{b,j}\|_M},$$

which equals their inner product after column-wise centering and unit-variance normalization. The *permutation alignment score* is

$$\text{Score}_{\text{perm}}(\mathbf{Y}_a, \mathbf{Y}_b) := \frac{1}{N} \max_{\Pi \in \mathbb{S}_N} \sum_{i=1}^{N} \rho(\mathbf{y}_{a,i}, \mathbf{y}_{b,\Pi(i)}),$$

where $\mathbb{S}_N$ is the set of permutations of $\{1, \dots, N\}$. (Our semi-matching score uses row-wise assignments rather than a bijection)

**Deflation Under Superposition**   Assume the columns of $\mathbf{Y}_a$ and $\mathbf{Y}_b$ are mean-centered and normalized to unit length. Under these conditions, the Pearson correlation between columns $\mathbf{y}_{a,i}$ and $\mathbf{y}_{b,j}$ simplifies to their dot product:

$$\rho(\mathbf{y}_{a,i}, \mathbf{y}_{b,j}) = \mathbf{y}_{a,i}^\top\mathbf{y}_{b,j}.$$

Define the cross-correlation matrix $\mathbf{G} = \mathbf{A}_a^\top\mathbf{A}_b$. Assuming the latent features are whitened, i.e., $\mathbf{Z}^\top\mathbf{Z} = \mathbf{I}$, we have:

$$\mathbf{Y}_a^\top\mathbf{Y}_b = (\mathbf{Z}\mathbf{A}_a)^\top(\mathbf{Z}\mathbf{A}_b) = \mathbf{A}_a^\top\mathbf{Z}^\top\mathbf{Z}\mathbf{A}_b = \mathbf{A}_a^\top\mathbf{A}_b = \mathbf{G}.$$

The permutation alignment score can then be expressed as:

$$\text{Score}_{\text{perm}}(\mathbf{Y}_a, \mathbf{Y}_b) = \frac{1}{N} \max_{\Pi \in S_N} \sum_{i=1}^{N} \mathbf{y}_{a,i}^\top \mathbf{y}_{b,\Pi(i)} = \frac{1}{N} \max_{\Pi \in S_N} \sum_{i=1}^{N} G_{i,\Pi(i)}.$$

Let $\mathbf{P}_\Pi$ be the permutation matrix associated with $\Pi$, where $[\mathbf{P}_\Pi]_{i,j} = 1$ if $j = \Pi(i)$ and 0 otherwise. Then, we can express the score as:

$$\text{Score}_{\text{perm}}(\mathbf{Y}_a, \mathbf{Y}_b) = \frac{1}{N} \max_{\Pi \in S_N} \text{tr}(\mathbf{G}\mathbf{P}_\Pi).$$

This formulation shows that the permutation score is determined by the matrix $\mathbf{G} = \mathbf{A}_a^\top \mathbf{A}_b$, which encodes the similarities between the mixing columns of the two systems. The score is the solution to a linear assignment problem, finding the permutation $\Pi$ that maximizes the sum of the diagonal entries of $\mathbf{G}$ after reordering its columns. From this formulation, it follows that the score is unity if and only if the columns of $\mathbf{A}_a$ and $\mathbf{A}_b$ are identical up to a permutation and sign flip. In all other scenarios, the score is deflated below one. This deflation can be characterized by the relationship between $\mathbf{A}_a$ and $\mathbf{A}_b$. We highlight two examples:

**(1) Different Sparsity Levels**    Suppose $\mathbf{A}_a = \mathbf{I}_F$ (each unit in system $a$ selects a single latent feature) and a column $j$ of $\mathbf{A}_b$ mixes exactly three features with equal weights. Let the support be $\{k, \ell, m\}$ and normalize the column to unit length: $\mathbf{A}_{b,\cdot j} = \frac{\mathbf{e}_k + \mathbf{e}_\ell + \mathbf{e}_m}{\sqrt{3}}$. Matching this unit to the corresponding single-feature unit in system $a$ (take $i = k$, so $\mathbf{A}_{a,\cdot i} = \mathbf{e}_k$) yields $G_{i,j} = \mathbf{e}_k^\top \mathbf{A}_{b,\cdot j} = \frac{1}{\sqrt{3}} \approx 0.58$. More generally, when one system uses pure single-feature columns while the other mixes exactly $n$ features equally per column (each column unit-normalized), the optimal pairing produces correlations that are systematically deflated by the factor $1/\sqrt{n}$.

**(2) Different Sparsity Patterns**    Consider sparse mixing matrices where each unit combines a small subset of features. If $\mathbf{A}_{a,i}$ has support on features $\{1, 2\}$ and $\mathbf{A}_{b,j}$ has support on features $\{2, 3\}$, then $G_{i,j} = \langle \mathbf{A}_{a,i}, \mathbf{A}_{b,j} \rangle$ reflects only the overlap in feature 2. The permutation algorithm must find pairs of units with maximal support overlap, but when feature usage patterns differ significantly between systems, this overlap is limited. For instance, suppose if systems $a$ and $b$ represent features $\{1, 2, 3, 4, 5, 6, \ldots\}$, but system $a$ groups them as $\{(1, 2), (3, 4), \ldots\}$ while system $b$ uses shifted groupings $\{(2, 3), (4, 5), \ldots\}$, each optimal pairing captures only partial feature overlap, leading to systematic deflation proportional to the degree of support mismatch.

**Perfect Alignment After Superposition Disentanglement**    Suppose each mixing matrix $\mathbf{A}_\bullet$ satisfies the $(K, \delta)$-restricted isometry property (RIP) [5] with $\delta < \sqrt{2} - 1$ and that an exact sparse recovery method is applied to recover the latent features from each observed representation. Then the recovered latent matrices are equivalent $\hat{\mathbf{Z}}_a \sim_\Pi \hat{\mathbf{Z}}_b \sim_\Pi \mathbf{Z}$ up to column permutations $\Pi$ and sign flips. Consequently,

$$\text{Score}_{\text{perm}}(\hat{\mathbf{Z}}_a, \hat{\mathbf{Z}}_b) = 1.$$

RIP ensures that each $K$-sparse latent vector $\mathbf{z}_m$ is the unique solution to the sparse recovery problem for each system. Since both representations are generated by the same latent matrix $\mathbf{Z}$, recovery in each system returns identical latent features (up to column reorderings). After column-wise normalization, perfectly matched columns yield correlation 1, giving an average score of 1.

This result establishes that disentanglement can achieve the theoretical ceiling for permutation alignment when the latent features are sufficiently sparse, the mixing matrices are well-conditioned, and an exact sparse recovery method is available.

## 4   Toy Models

We next explore the use of toy models to see if alignment interference and recovery occurs in practice. The experiment proceeds as follows: (1) models are trained with different seeds to induce distinct superposition arrangements, (2) SAEs are trained for each model's activations separately to disentangle superposition, and finally (3) we observe whether predictive mapping alignment scores over SAE latents achieve higher alignment than those over base neurons. We follow the methodology of [13] to train toy models which exhibit superposition.

## 4.1 Experimental Setup

**Feature Dataset** We generate ground-truth sparse feature data $\mathbf{Z} \in \mathbb{R}^{M \times F}$, where each of the $M$ datapoints is a $F$-dimensional vector of features. Each feature has a 10% probability to be a real number uniformly sampled in the range of $[0, 1)$, and is 0 otherwise. A task importance vector $\mathbf{T} \in \mathbb{R}^F$ is defined so that each feature has a unique weight for its reconstruction error. The values are sampled from a power-law of $\frac{1}{x^2}$ at $F$ uniformly spaced intervals between $x = [1, 3 - \frac{2}{F}]$. This is done to reflect the distribution of features in a natural domain such as vision.

**Model Architecture and Training** Each toy model is a two-layer autoencoder with tied weights, $\hat{\mathbf{z}} = \mathbf{W} \operatorname{ReLU}(\mathbf{W}^\top \mathbf{z}) + \mathbf{b}_{dec}$, where $\mathbf{W} \in \mathbb{R}^{F \times N}$ with $N < F$. Training minimizes the task importance-weighted reconstruction loss, $\mathcal{L} = \frac{1}{B} \sum_{i=1}^{B} \mathbf{T}^\top (\mathbf{z}_i - \hat{\mathbf{z}}_i)^2$. Toy models are trained for one epoch with a batch size $B = 1024$. A pair of $N$-neuron models are initialized with different seeds to induce distinct superposition arrangements after training. Superposition of features is encouraged from the sparsity of the features and there being less neurons than the number of features to represent.

**Interpreting Toy Models** An advantage of these toy models is how readily interpretable they are: the learned weights $\mathbf{W}$ directly map features to neurons, permitting us to inspect which features are represented and how they are arranged in superposition. This is important as we want to first confirm that different arrangements can occur in practice, at least in toy models.

To measure if a feature is represented, we simply take the magnitude of that feature's weights across the neurons: $\|\mathbf{W}_i\|_2$ for the $i^{\text{th}}$ feature. As per [13], a norm of $\approx 1$ indicates it is represented by the model, and the model does not represent features with norms significantly below 1. We wish to identify the features represented in both seeded models, so we multiply their respective norms together. Specifically, we consider a feature $i$ to be represented in both seeded models 1 and 2 when: $\|\mathbf{W}_i^{(1)}\|_2 \times \|\mathbf{W}_i^{(2)}\|_2 \geq 1$. While this threshold is somewhat arbitrary and serves more as a heuristic, it provides a practical way to focus our analysis on features that are well-represented in both models.

Once the shared features are identified, we proceed to compare their superposition arrangements across the seeded models. The rows of both models' $\mathbf{W}$ are indexed by the shared features to obtain $\mathbf{W_s}$. The $i^{\text{th}}$ column of a model's $\mathbf{W_s}$ is how all the shared features are allocated for the $i^{\text{th}}$ neuron. Semi-matching is taken across the columns of the seeded models' respective $\mathbf{W_s}$. Doing so finds the best superposition arrangement match for the shared features between the neurons of the models. The lower the neurons' maximum matching scores are, the more indicative that the shared features are superimposed in different arrangements across the two models.

**SAE Architecture and Training** We train TopK SAEs [15, 38] to disentangle features from superposition, setting the number of latents to $F$ to match the number of features. An SAE reconstructs neural activations by first encoding into a sparse overcomplete space, followed by a decoding into the predicted reconstructions. More SAE details are provided in the appendix A.2.

**Validation of Superposition Disentanglement** To validate if our SAEs disentangle the features from superposition, we compute semi-matching between the ground-truth feature values in our dataset and (1) the toy model's base neural activations to the features, (2) the subsequent SAE latent activations from passing the base activations through the SAE encoder. Results are presented in the appendix A.3, where we see SAE latents consistently have a better match with feature values compared to neurons across all models. This suggests that for the features represented by the toy models, SAEs mostly succeed in inverting the toy model, thus recovering the original feature values.

**Alignment Scores** Alignment scores are taken across the differently seeded models of the same size. 20% of the feature dataset is held out of SAE training to compute the alignment scores. This subset is fed to the toy models, where their neural activations are in turn fed to the SAEs to obtain their latent activations. The latent activations from the encoder have the TopK activation applied followed by a ReLU. We first iterate through the folds and mark the latents which did not fire for a single stimulus in the fold. After accumulating these dead latents across folds, we delete all of them from the data before reiterating over the folds to compute the alignment. To control for alignment improvements that might arise solely from increased dimensionality and sparsity, we use randomly initialized SAEs with identical architecture as baselines.

## 4.2 Results

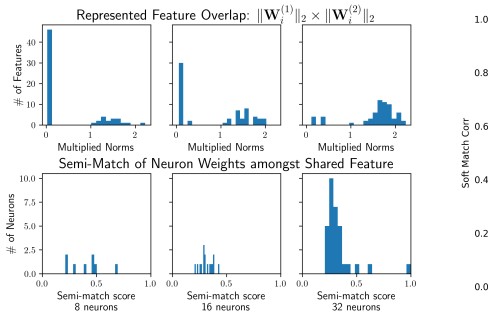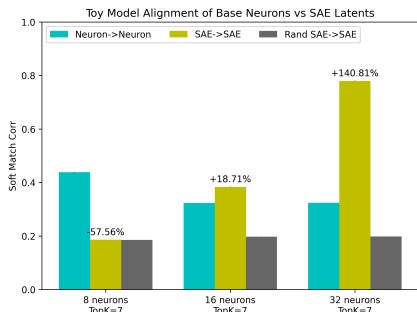

Figure 2: Feature overlap (left, top row) and the superposition arrangement comparison (left, bottom row) of the shared features (multiplied norms $\geq 1$) between the differently seeded toy models ($N = 8, 16, 32$ models are displayed along columns). Soft-matching alignment (right) sees a significant increase when SAE latents are replaced with base neurons for $N = 16$ and $32$.

**Superposition Arrangement Comparison** For our experiments, we generate a feature dataset with $M = 10,240,000$ and $F = 64$ and train a pair of toy models with different initializations across three different sizes for $N$: 8, 16, and 32. Presented in the left grid of plots in Figure 2 are the shared feature overlap (top row) and superposition arrangement (bottom row) results for each pair. We see the distribution of shared features across the different seeds, increasing as $N$ grows due to a higher capacity for the models to represent all the features. Importantly, the bottom row shows that for the features shared across the seeds, they are arranged across the model's neurons in different ways. Almost all the neurons have a max semi-matching correlation of $< 0.5$, demonstrating that the same features take different superposition arrangements in otherwise identical models.

**Alignment Results** The right plot in Figure 2 shows soft-matching correlation scores (mean across k=5 folds with error bars) between the differently seeded models over: (1) their respective neural activations (Neuron→Neuron), (2) their SAE latent activations (SAE→SAE), and (3) the randomly initialized SAE activations to the same trained latent activations from the other model (Rand SAE→SAE). There is a large alignment increase for SAE→SAE versus Neuron→Neuron in the 16 and 32 neuron conditions. For the decrease in the 8 neuron condition, we suspect this may be due to the low shared feature overlap between the two seeds. In the 32 neuron condition, both models represent nearly all 64 features, so it is reassuring to see the alignment approach the ground-truth of 1. Why it falls short may be due to the residual reconstruction errors of the SAEs, and also fundamental limitations in what they can disentangle [40]. Semi-matching alignment is also computed, with results found in the appendix A.4 due to high consistency with soft-matching.

## 5 Real Models

How well do these toy model findings generalize to full-scale DNNs? The challenge is that real models lack identifiable ground-truth features or interpretable superposition arrangements, making direct validation impossible. We therefore test our predictions indirectly: if superposition causes alignment deflation in real networks, then disentanglement with SAEs should improve alignment scores between corresponding layers of independently trained models with identical architectures.

**Experimental Setup** We evaluate our findings on ImageNet-trained DNNs [10] using two architectures: ResNet50 [17] and a vision transformer ViT-B/16 [12]. For each architecture, we train pairs of models with different random seeds, hypothesizing they will learn overlapping features in different superposition arrangements consistent with toy models. TopK SAEs are trained on neural activations to the ImageNet training set from an intermediate layer (layer2.3.bn2 for ResNet50, layer5 MLP block output for ViT) and a deep layer (layer4.2.bn2 for ResNet50, layer11 MLP block output for ViT) for both seeds. For ResNet50, we extract activations from the center-channel position; for ViT, we use the `cls` token position. Soft-matching is computed for a given layer between the different seeded models using 50,000 ImageNet validation images, with dead latents removed ($< \approx 3\%$).

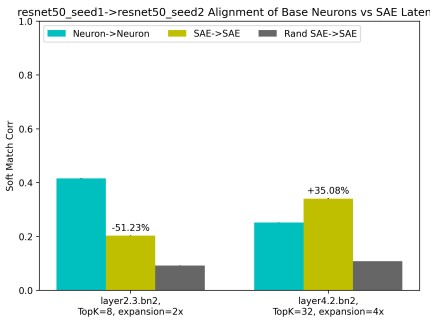 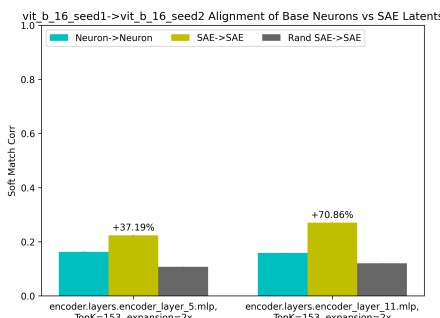

Figure 3: Soft-matching alignment between differently seeded DNNs (left: ResNet50, right: ViT-B/16) trained on ImageNet object classification.

**Results** Figure 3 shows soft-matching correlation scores for SAE latents versus base neurons across model pairs. The results reveal depth-dependent patterns that show deflated alignment under superposition. ResNet50 exhibits a clear transition: SAE disentanglement decreases alignment at the intermediate layer (layer2.3.bn2) but increases alignment at the deeper layer (layer4.2.bn2). ViT shows consistent improvements at both depths, with larger gains at the deeper layer (layer11 vs. layer5). The alignment increases demonstrate that superposition actively obscures true feature correspondences in real neural networks. When SAEs successfully disentangle these superimposed features, the underlying representational similarities between independently trained models become visible. The depth-dependent pattern aligns with the superposition hypothesis: deeper layers process increasingly complex, sparser features which are more conducive to superposition. Intermediate layers, particularly in ResNet50, may contain simpler features that are already sufficiently disentangled, making SAE reconstruction artifacts more harmful than beneficial.

# 6 Linear Regression and DNN→Brain Alignment

Do our results translate to linear regression, the most flexible of the mapping metrics? And do they extend to alignment between DNNs and the brain? These are the most relevant questions for NeuroAI and we address them in this section.

**Experimental Overview** We compute ridge regression correlation scores for the same toy model and seeded-DNN activations, sweeping over integers in $[-8, 8]$ for the optimal regularization coefficient $\alpha$. Crucially, linear regression only requires the source model to be disentangled, as the linear map can handle remixing features into the target's superposition arrangement in its base neurons. In particular, we want to predict target neurons $Y_b = ZA_b$ from either source neurons $Y_a$ or SAE latents $\hat{Z}_a = SAE(Y_a)$. If the SAE approximately disentangles the true features up to permutations $\Pi$, then the optimal regression coefficients $\min_{W^*} \|Y_b - \hat{Z}_a W^*\|$ would simply be $W^* = \Pi A_b$. As a consequence, we focus on SAE→Neuron scores in this section, with the random SAE control also being mapped onto the base neurons rather than the trained SAE latents of the target.

This linear mixing is ideal for DNN→brain alignment, where limited voxel data makes training reliable brain SAEs impractical. For brain data, we use the Natural Scenes Dataset (NSD) [1], a large-scale fMRI dataset of humans viewing natural images. We analyze high-level ventral stream voxel responses from the four subjects who viewed each of 10,000 images across three repetitions (more details found in appendix A.5). DNN activations used to align with the brain are from the penultimate layers of ResNet50 (layer4.2) and ViT-B/16 (encoder_layer_11) (same positional extraction as last section), using the default pretrained torchvision weights to facilitate replication. Alignment scores are normalized by dividing raw scores by the mean subject-wise noise ceiling across voxels.

**Results** As shown in Figure 4, the SAE→Neuron regression scores are generally higher than the Neuron→Neuron scores for the same toy models (top) and DNNs (bottom) previously studied. This finding demonstrates that disentangling the source network significantly improves our ability to predict activations in a target network, a result with important implications for cross-model generalization and interpretability. If superposition obscures shared computational patterns between

networks, then disentanglement should reveal these hidden correspondences, enabling more accurate linear mappings between representations that process the same underlying information.

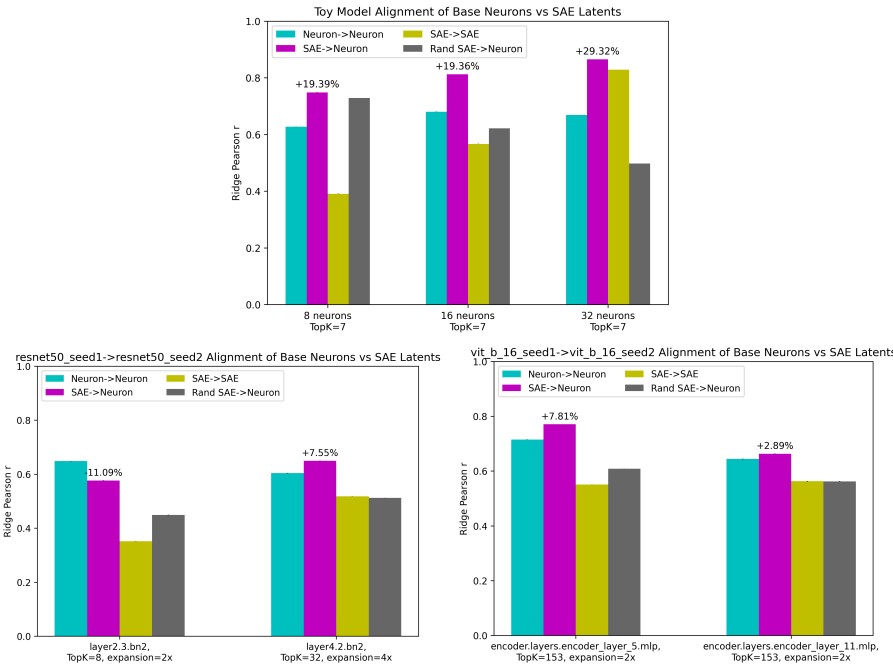

Figure 4: Regression correlation scores between the same toy models (top) and DNNs (bottom).

However, the SAE→SAE scores present a puzzling pattern: they consistently underperform Neuron→Neuron baselines, with only the N=32 toy model showing improvement. This systematic decrease suggests that applying SAEs to both source and target introduces cumulative reconstruction errors that outweigh the benefits of disentanglement. Two factors likely contribute to this effect: first, compounding reconstruction noise from two imperfect SAE reconstructions may amplify artifacts that disrupt linear relationships; second, the expanded overcomplete feature spaces from both SAEs may introduce spurious dimensions and redundancies that complicate regression, unlike the asymmetric SAE→Neuron case where the target space remains constrained.

Interestingly, this contrasts with our soft-matching results where SAE→SAE typically improved alignment. However, this issue may also affect permutation-based metrics, the key difference being that the benefits of disentanglement outweigh reconstruction costs when baseline alignment is severely deflated by superposition. In other words, while SAE imperfections harm both metrics, permutation-based methods see net gains because superimposed representations have such poor one-to-one correspondences that even imperfect disentanglement represents a substantial improvement.

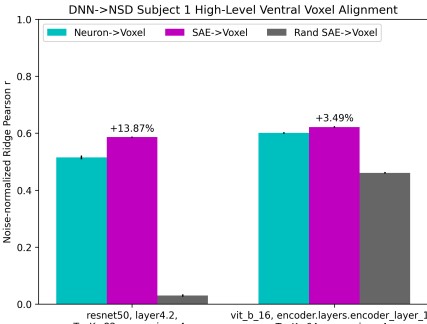

Figure 5: Regression correlation scores (normalized by subject's mean voxel noise ceiling) between DNNs and the human ventral visual cortex using NSD voxel responses from subject 1. Both ResNet50 (left) and ViT (right) see increased alignment from superposition disentanglement.

We next examine whether these effects extend to DNN-brain alignment. Figure 5 reveals notable improvements in DNN-brain alignment when DNN representations are disentangled. ResNet50

shows a 13.9% increase in linear predictivity of brain voxel responses, while ViT achieves a 3.5% improvement. These gains are meaningful in NeuroAI, where even modest improvements in brain prediction accuracy are significant given the inherent noise and complexity of neural data. If superposition limits how well DNNs align with biological vision, then disentanglement may be uncovering computational principles shared between artificial and biological systems. The fact that SAE latents, trained purely on DNN activations without any brain data, improve brain prediction suggests that the features recovered through disentanglement are more neurally plausible than the original superimposed representations. We report alignment scores for one representative subject, with the remaining three subjects showing highly consistent patterns (see Appendix A.6).

## 7    Discussion

**Limitations**    While results for DNN→DNN and DNN→brain alignment are consistent with the toy models, we cannot verify the superposition arrangement of the DNN features directly. It therefore remains possible that other factors might contribute to the alignment increase, such as the SAE constructing new representations rather than superposition disentanglement per se.

Our approach inherits known SAE limitations, including feature consistency, absorption, and identifiability problems [41, 33, 6, 40]. We did not exhaustively search through SAE architectures which may alleviate these concerns [14, 4]. Despite this, we emphasize that our hypothesis is agnostic to the exact dictionary learning approach; we adopt SAEs as it is currently the most effective means to disentangle superposition. If new and improved methods come along, it would be of great interest to repeat these experiments with them.

Lastly, we only provide limited DNN→DNN and DNN→brain alignment results. Scaling to diverse architectures, cross-architectural comparisons, additional brain datasets, and other domains like language processing remain important future work. Also, extending theory to encompass linear regression and neural geometry metrics like RSA is needed to put our observations on firmer footing.

**Future Directions**    The superposition hypothesis has fundamentally reshaped MI research, revealing that disentanglement can be beneficial to extract features from superimposed arrangements that would otherwise remain hidden. Here we demonstrate that superposition also has transformative implications for representational alignment research. Our results show that different superposition arrangements can systematically deflate alignment scores between networks that learn identical underlying features, creating spurious dissimilarity. The implications extend beyond methodological concerns. If superposition arrangements vary systematically, due to architectural differences, training procedures, or stochastic factors, then our results suggest that many established findings about representational similarity, model comparisons, and brain-AI alignment may require reexamination. To speculate how this could impact previous studies, the diminishing returns of brain predictivity as a function of task performance [37] might be due to increased superposition in the highest-performing models. Or perhaps inconsistencies among alignment metrics [47] is partly due to how these metrics are differently impacted by superposition. Our work may encourage the wide-spread adoption of dictionary learning in NeuroAI to re-assess these findings and generate new ones.

Understanding the forces acting on superposition arrangements is another key direction. Different superposition arrangements seem to occur in toy models, but it is unclear if this translates to full-scale DNNs or brains. A recent study indicates that DNNs and brains possess "privileged axes" [25], neurons/voxels with similar tuning curves across models or conspecifics. As these axes need not be interpretable, it could be that neural networks converge onto similar superposition arrangements. Relatedly, it is uncertain that brains exhibit superposition at all, but studies which find interpretable population codes in brain data via dictionary learning [24, 26] offer some indications. More scrutiny should be devoted to the nature of superposition arrangements; testing the forces of feature sparsity, co-occurrence, and task importance on toy models may reveal insights which transfer to real models.

Lastly, there is perhaps a positive externality in taking representational alignment over SAE latents. Doing so may inject interpretability into alignment research, providing a more qualitative understanding of which features drive DNN-brain representational alignment.

Altogether, this convergence of superposition theory with representational alignment opens new theoretical and empirical frontiers that may significantly alter how we understand and compare neural representations across artificial and biological neural networks.

## Acknowledgments

The authors are appreciative of Liv Gorton, Shaan Shah, and Habon Issa for encouraging discussions during the nascent stage of this effort.

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

# A Technical Appendices and Supplementary Material

## A.1 Semi-matching Score

The semi-matching (also known as pairwise) score quantifies direct correspondence between units across two representations [25, 35]. Suppose we have response matrices $\mathbf{Y}_a \in \mathbb{R}^{M \times N_a}$ and $\mathbf{Y}_b \in \mathbb{R}^{M \times N_b}$ containing the activity of $N_a$ and $N_b$ units respectively for $M$ stimuli. The metric proceeds in two stages:

1. **Assignment.** For each unit in $\mathbf{Y}_a$, we find its most correlated partner in $\mathbf{Y}_b$ using a bipartite semi-matching:

$$A^* = \max_A \sum_{i=1}^{N_a} \sum_{j=1}^{N_b} \rho(\mathbf{Y}_a^{\text{train}}{}_i, \mathbf{Y}_b^{\text{train}}{}_j) A_{ij} \qquad (2)$$

$$\text{s.t. } A_{ij} \in \{0,1\}, \quad \sum_j A_{ij} = 1.$$

   This ensures every unit in $\mathbf{Y}_a$ is matched to exactly one unit in $\mathbf{Y}_b$, though a single unit in $\mathbf{Y}_b$ may be matched to multiple units in $\mathbf{Y}_a$.

2. **Evaluation.** Using these assignments, we measure correlations on held-out test data and average across pairs:

$$s_{\text{semi}}(\mathbf{Y}_a^{\text{test}}, \mathbf{Y}_b^{\text{test}}, A^*) = \frac{1}{N_a} \sum_{i=1}^{N_a} \sum_{j=1}^{N_b} \rho(\mathbf{Y}_a^{\text{test}}{}_i, \mathbf{Y}_b^{\text{test}}{}_j) A_{ij}^*. \qquad (3)$$

This metric is asymmetric by design since it depends on the choice of source versus target. The semi-matching formulation is useful when the two systems differ in size, allowing for units in the larger system to remain unmatched.

## A.2 Sparse Autoencoder Details

**TopK SAE Architecture** Activations from $N$ neurons, $\mathbf{x} \in \mathbb{R}^N$, are extracted from a population (e.g., a layer) in response to a stimulus. A TopK sparse autoencoder (SAE) [15] linearly encodes these neural activations and performs a TopK activation function to produce a sparse overcomplete code $\mathbf{z} \in \mathbb{R}^F$, where $F > N$. The TopK activation function leaves the $k$ highest values in $\mathbf{z}$ as is, and sets all other values to 0. Then $\mathbf{z}$ is linearly decoded to produce $\hat{\mathbf{x}} \in \mathbb{R}^N$, an estimated reconstruction of $\mathbf{x}$. This can all be expressed as:

$$\mathbf{z} = \text{TopK}(\mathbf{W}_{enc}\mathbf{x} + \mathbf{b}_{enc})$$

$$\hat{\mathbf{x}} = \mathbf{W}_{dec}\mathbf{z} + \mathbf{b}_{dec}$$

where $\mathbf{W}_{enc} \in \mathbb{R}^{F \times N}$, $\mathbf{b}_{enc} \in \mathbb{R}^F$, $\mathbf{W}_{dec} \in \mathbb{R}^{N \times F}$, and $\mathbf{b}_{dec} \in \mathbb{R}^N$ are all learned parameters.

**SAE Training** The SAE is trained to reconstruct the neural activations, together with an auxiliary loss to prevent "dead latents", values in $\mathbf{z}$ which remain $0$ for $t$ gradient descent steps. Toy model SAEs are trained for one epoch on 80% of the generated feature dataset $\mathbf{Z}$, and DNNs are trained on activations to the ImageNet training set for 300 epochs. The loss function is the mean squared error of reconstruction summed with the auxiliary loss: $\mathcal{L} = \mathcal{L}_{mse} + \alpha_{aux}\mathcal{L}_{aux}$ where

$$\mathcal{L}_{mse} = \frac{1}{B}\sum_{i=1}^{B}\left\|\mathbf{x}_i - \hat{\mathbf{x}}_i\right\|_2^2$$

$$\mathcal{L}_{aux} = \frac{1}{B}\sum_{i=1}^{B}\left\|\mathbf{e}_i - \hat{\mathbf{e}}_i\right\|_2^2, \qquad \mathbf{e} = \mathbf{x} - \hat{\mathbf{x}}, \qquad \hat{\mathbf{e}} = \mathbf{W}_{dec}\mathbf{z}_{dead} + \mathbf{b}_{dec}$$

$B$ is the batch size, $\alpha_{aux}$ is a scaling factor, and $\mathbf{z}_{dead}$ contains the top-$k_{\text{aux}}$ pre-TopK activation values of dead latents (live latents are set to $0$). In our experiments, we set the learning rate to 1e-3 for toy models (4e-4 for DNNs), batch size to 1024, dead steps $t$ to 1 for toy models (32 for DNNs), $\alpha_{aux}$ to 1e-1 for toy models (1e-4 for DNNs), and $k_{\text{aux}}$ to $F$. For DNNs, $F$ is set to $N$ multiplied by an expansion factor specified in the figures.

### A.3 Superposition Disentanglement in Toy Models

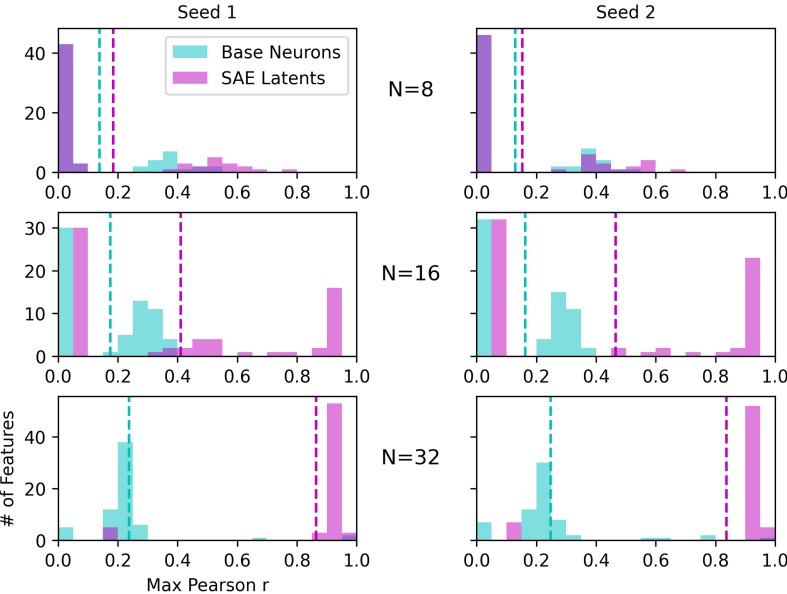

Figure 6: Semi-matching correlations between features and toy model neurons (cyan) and SAE latents (magenta). Vertical dashed lines are the respective means across features, where the SAE latents have higher mean correlations over neurons across all models.

A validation is provided that our toy model SAEs indeed disentangle superposition. Each feature's ground-truth values are taken from the same 20% of the generated dataset used for computing alignment. Semi-matching is computing between these values and a toy model's neural activations to this subset, where the maximum correlation among neurons is recorded. This is then repeated between feature values and SAE latent activations. Each feature's maximum correlation for the neurons and SAE latents for all models are plotted on a histogram in Figure 6. It's shown that SAE latent activations have consistently higher correlations with the features (mean across features plotted as a vertical dashed line), meaning they more closely resemble the original features relative to neurons, disentangling them from superposition. We also note that the separations between neuron and SAE latent distributions become more pronounced as the model size increases due to their increased

capacity to represent more features. This is because unrepresented features will not be recovered by the SAE, driving both neuron and SAE latent correlations to those features to 0.

## A.4 Semi-matching Results

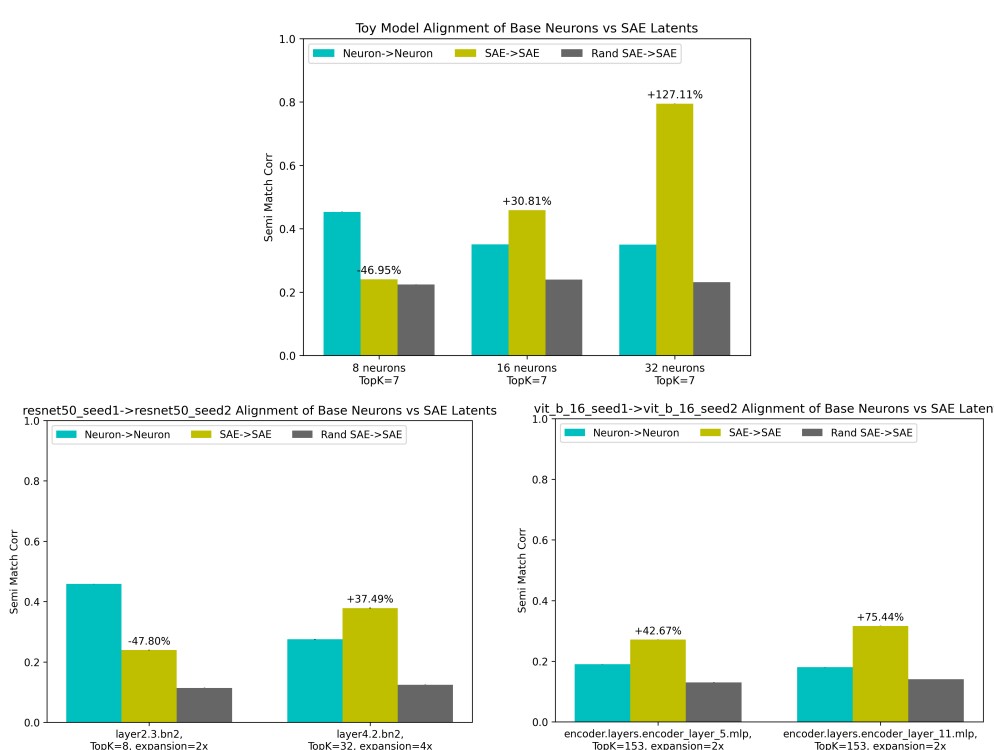

Figure 7: Semi-matching alignment between the same toy models (top) and DNNs (bottom).

## A.5 Natural Scenes Dataset Details

A detailed description of the Natural Scenes Dataset (NSD; http://naturalscenesdataset.org) is provided elsewhere [1]. Here, we briefly summarize the data acquisition and preprocessing steps. The NSD dataset contains measurements of fMRI responses from 8 participants who each viewed 9,000–10,000 distinct color natural scenes (22,000–30,000 trials) over the course of 30–40 scan sessions. Scanning was conducted at 7T using whole-brain gradient-echo EPI at 1.8-mm resolution and 1.6-s repetition time. Images were taken from the Microsoft Common Objects in Context (COCO) database [36], square cropped, and presented at a size of 8.4° x 8.4°. A special set of 1,000 images were shared across subjects; the remaining images were mutually exclusive across subjects. Images were presented for 3 s with 1-s gaps in between images. Subjects fixated centrally and performed a recognition task in which they were instructed to indicate whether they have seen each presented image at any point in the past. Informed consent was obtained from the participants and the study was approved by the University of Minnesota Institutional Review Board.

The fMRI data were pre-processed by performing one temporal interpolation (to correct for slice time differences) and one spatial interpolation (to correct for head motion). A general linear model was then used to estimate single-trial beta weights, with the HRF estimated for each voxel and the GLMdenoise technique used for denoising [20, 42]. In this work, we used the 1.8mm volume 'nativesurface' preparation of the NSD data and version 3 of the NSD single-trial betas ('beta fithrf GLMdenoise RR'). Every stimulus considered in this study had 3 repetitions, and we only analyzed data in the four NSD subjects who had full 3 repetitions for each of their 10,000 images. To derive voxel responses to each stimulus we averaged single-trial betas after z-scoring every voxel within each scan session.

We selected ventral visual stream voxels by using the "streams" atlas provided in the native space of each subject with NSD [35]. Briefly, this ROI collection reflects large-scale divisions of the visual cortex into primary visual cortex and intermediate and high-level ventral, lateral and dorsal visual areas. These were manually drawn for each subject by NSD curators and were based on voxel-level reliability metrics. For this study, we extracted the ROI mask corresponding to the 'higher-level ventral stream' label. This ROI was drawn to follow the anterior lingual sulcus (ALS), including the anterior lingual gyrus (ALG) on its inferior border and to follow the inferior lip of the inferior temporal sulcus (ITS) on its superior border. The anterior border was drawn based on the midpoint of the occipital temporal sulcus (OTS). This parcel is very broad, and results in 7,000-9,000 voxels per subject.

## A.6 Ridge Regression Results for Remaining NSD Subjects

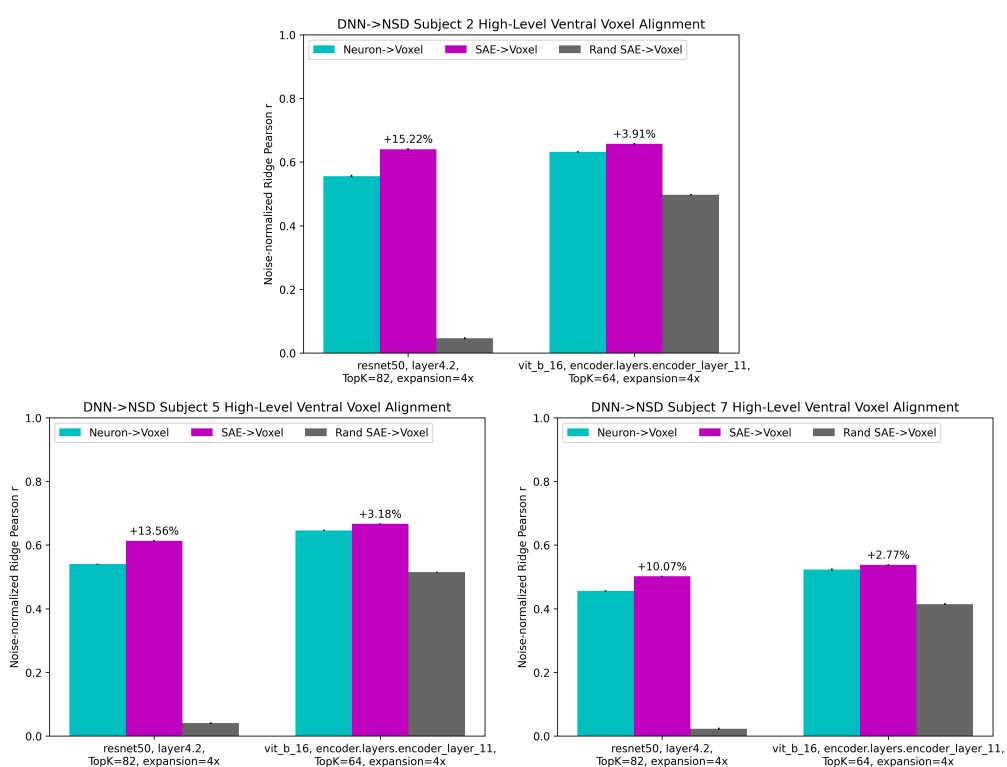

Figure 8: Ridge regression alignment (adjusted by subject's mean voxel noise ceiling) for the remaining subjects 2 (top), 5, and 7 (bottom). The same neural and SAE activations are used across all NSD subjects.

