# OpenReview forum: "Superposition disentanglement of neural representations reveals hidden alignment"
_NeurIPS.cc/2025/Workshop/UniReps — UniReps2025 oral_

### Official Review · Reviewer_8HA1 · 2025-09-11
**The paper is well-written and tackles an interesting question on representation alignment under superposition hypothesis, but its claims about SAE disentanglement lack empirical support and key experimental details are missing.**

**Confidence:** 3

**Review:**

### Summary
The paper is based on a theoretical foundation and argues that strict permutation-based metrics depend on superposition arrangements. This leads to lower alignment between models containing superposition neurons. To support this hypothesis, the authors demonstrate improved alignment when comparing SAE latents, which are assumed to be more disentangled than baseline neuron representations.

### Strengths
- The paper is well-written and clearly structured.
- It addresses an interesting question about the alignment between model-learned representations and the brain.
- The theoretical explanation of the alignment score is thorough and easy to follow.


### Weaknesses
- The central hypothesis is that SAE representations are sparser and can better disentangle the feature space compared to baseline neurons, which is also a common assumption in related work. However, the disentanglement ability of SAE latents remains somewhat vague, as it strongly depends on the training process. The paper’s argument would be stronger if the authors provided a concrete measurement of the mono-semanticity of SAE latents in their toy experiments.

- In section 6, the authors suggest that the failure of SAE -> SAE in linear regression may be due to imperfections in training the SAE latents. However, this claim is not empirically supported (e.g., no reconstruction loss curves or related metrics from SAE training are provided)

- Some details on hyperparameter choices are missing.

### Questions
- What is the dimensionality of SAEs used in real models? How do the authors define K in TopK SAE training?

- Since you already propose a metric to measure superposition arrangements, do you also apply it to SAE representations? I am particularly curious about how this relates to the mono-semantic properties of SAE latents in your experiments.

**Score:**

2

**Topic Fit:**

3

---

### Official Review · Reviewer_SWts · 2025-09-14
**Meaningful hypothesis but the design and experiments need strengthening.**

**Confidence:** 4

**Review:**

**Summary**:

This work explores whether superposition has an adverse effect on representational alignment metrics, i.e. does considering "features", rather than neurons, highlight the underlying alignment between two representations better? By way of background, the authors recapitulate the soft-matching correlation score and the permutation alignment score. The authors factorize representations $\mathbf{Y}$ as a product of a latent matrix $\mathbf{Z}$ and a mixing matrix $\mathbf{A}$.  They state a proposition demonstrating the conditions required for perfect alignment in latent factors (under the latter metric). Under the assumption that two systems learn the same latent matrix, they demonstrate that a misalignment between mixing matrices can lead to "deflation" in alignment scores. In toy models the authors find that, given enough representational capacity in original model, disentangled representations have a higher alignment than comparing raw neurons and a random baseline. In real models, they find that SAE->SAE alignment is improved for the soft matching score but not for the regression metric.

 I would be curious what the authors think about crosscoders [1] (shared encoder, per-system linear decoders) and how those fit in within their hypothesis? For two systems, crosscoders model the $Y_{a}=ZA_{a}$ and $Y_{b}=ZA_{b}$ decomposition, such that $Z$ is shared between systems by construction.

**Strengths**:

The authors are working with a sound and interesting hypothesis: if SAEs really do provide disentanglement, then it would be reasonable for differently seeded models (under the assumption that they learn similar representations) to demonstrate increased alignment in feature space. The authors also make an effort towards some statistical hygiene (cross validation and random baselines).

**Weaknesses**:

Section 4: in the toy model, the authors consider a feature $i$ to be represented iff $\lVert W_{i}^{1}\rVert \times \lVert W_{i}^{2}\rVert\geq1$. Additionally, the authors claim that a row norm of  $\approx 1$ indicates that a feature is represented by the model.
1. Norm $\approx 1$ is  not substantiated, why is this the threshold? Is this empirically selected? The toy model does not constraint norms. The next two points assume this requirement.
2. For any feature $i$, a large norm in the first seed and a small norm in the second seed can still pass the threshold of the product. Under the first requirement, this is a false positive.
3. For any feature $i$, a $0.99$ norm in both seeds will not satisfy the threshold of the product. Under the first requirement, this is a false negative.

I recommend explaining the norm threshold and replacing the product with a minimum.

Section 5:
1. The lower alignment in SAE space for layer2 ResNet 50 is not given enough treatment. Does this decreased alignment happen with a higher capacity (i.e. expansion factor) SAEs? And, the narrative carrying over from the toy model suggests that neuron-neuron metrics "undersell" the alignment. Why is it that some cases "oversell" the alignment?
2. The authors claim a depth-dependent pattern, but only presents results from two layers in both models. Additionally, in the case of ResNet50, the SAE configurations are different (see x-axis for Figure 3 left.) making the depth-dependent claim more shaky.

I recommend conducting this analysis with more layers per model and at a few more SAE configurations.

Section 6:
1. It seems that SAE-SAE regression correlation paints a different picture than SAE-SAE soft matching alignment. The authors acknowledge that this puzzling. However, they explain this with another hypothesis, offering a plausible but under-substantiated answer to conflicting evidence.

Overall, I would also recommend the authors tie their experimental set up with the framework they have set up in the Sections 2 and 3. From a communication perspective, Sections 2 and 3 do not have a clear enough connection to Sections 4-6.

[1] : https://transformer-circuits.pub/2024/crosscoders/index.html

**Score:**

3

**Topic Fit:**

3

---

### Official Review · Reviewer_fris · 2025-09-14
**Nice theoretical study that demonstrates the failure of representational alignment to account for superposed codes**

**Confidence:** 4

**Review:**

Nice theoretical study that demonstrates that similar information in superposition (rather than by selective neurons) fails to be captured by standard representational alignment measures. Disentangling this effect is important for comparing artificial and biological neural systems, which often represent information in superposition. A clear accept for the workshop.

Minor: You may be interested in past failures of representational alignment frameworks, uncited here:

* CKA fails to recover same-architecture inits ([Han et al., 2023](https://proceedings.mlr.press/v202/han23d))
* RSA recovers ground-truth geometries ([Schütt et al., 2023](https://elifesciences.org/articles/82566))
* RSA etc. fail to achieve high sensitivity/specificity in recovery of functional similarity i.a. ([Klabunde et al., 2023](https://arxiv.org/abs/2305.06329); [Ding et al., 2021](https://arxiv.org/abs/2108.01661); [Bo et al., 2024](https://arxiv.org/abs/2411.14633))
* And the classic [Jonas & Kording (2017)](https://doi.org/10.1371/journal.pcbi.1005268)

**Score:**

4

**Topic Fit:**

3

---

### Official Review · Reviewer_tsNz · 2025-09-15
**An interesting direction to use SAEs to compare the learned disentangled representations of networks, rather than the superimposed concepts within individual neurons, for evaluating network similarity and representation alignment.**

**Confidence:** 5

**Review:**

They study whether the change in the superposition can impact the metrics being used to study the alignments with neural representations. They show that instead of comparing the neural representations, one can have an increase in the alignment score when the sparse representations (that disentangle the superposition) of the neural representations, achieved by SAEs. Hence, suggesting to uncover representational alignment via mechanistic interpretability frameworks such as SAEs.


On quality, clarity, and originality: This paper is very well-organized, and very well-written. I'm not fully an expert in representation alignment, but do know well about MI and SAEs. The angle on using SAEs, and that superposition impact the alignment metrics seem new and very good observation. Lastly, this is among the first papers combining representation alignment with mechanistic interpretability (SAEs). Result sections writing needs improvement.

I enjoyed reading the paper: this is the first paper I see that explain the superposition hypothesis with such a simple logic and description. Well-done.

Suggestions/Questions:

- My impression is that using Topk-SAEs and comparing the sparse representations of the internal representations at shallow depth does not properly reflect alignment. Please discuss how this can be addressed; could it be that the architectural design of SAE is limited to only disentangle concepts at deep layers? Then, any suggestions on how to compare models at early layers?

- SAEs might be biased toward certain features regardless of the data DNNs are trained on. Existence of such bias (if true) can be a source of increase in representation alignment regardless of closeness of networks. Additional experiment is recommended to use SAEs for an analysis on "difference" and dissimilarity between networks.

- Strongly recommended to include other representational alignment metrics (those not sensitive to individual neurons in the figure). How is the gain of SAEs->SAEs compared to neuron-neuron for those metrics?

- Expand the result section exploring other alignment metrics, and other SAEs.

Limitations:
- While the paper has mentioned their focus is on alignment metrics that evaluate similarity at individual neuron level, studying the question of the paper on more sophisticated metrics that are not sensitive to representation form is strongly recommended. Otherwise, your statement as "necessary" would be weak.
- Result section organization can be improved.

minor:

- Popular SAEs use untied encoder/decoder. Please clarify why the used SAE has tied weights.

- line 65: "necessary"? this is strong. your results suggest "significantly useful".

- for RIP, cite [1].

- line 109: please be more precise on "well-conditioned", and "an exact sparse recovery method is available". For example, SAE always ideal?

- line 112: cite or define Pearson correlation.

- suggest to include a visualization of the data from the toy model.

- please also cite the original TopK AE [2].

[1] Candes, E. J. (2008). The restricted isometry property and its implications for compressed sensing. Comptes rendus. Mathematique, 346(9-10), 589-592.

[2] Makhzani, A., & Frey, B. (2013). K-sparse autoencoders. arXiv preprint arXiv:1312.5663.

**Score:**

4

**Topic Fit:**

3